# Exploring cooking fuel choices among Ghanaian women of reproductive age: A socio-economic analysis from a statistical mechanics perspective

**Richard Kwame Ansah**[1,2]*, **Richard Kena Boadi**[2], **William Obeng-Denteh**[2], **Killian Asampana Asosega**[2,3], **Kassim Tawiah**[1,3]

1 Department of Mathematics and Statistics, University of Energy and Natural Resources, Sunyani, Ghana,
2 Department of Mathematics, Kwame Nkrumah University of Science and Technology, Kumasi, Ghana,
3 Department of Statistics and Actuarial Science, Kwame Nkrumah University of Science and Technology, Kumasi, Ghana

☯ All these authors are contributed equally to this work
* richard.ansah@uenr.edu.gh

**Data Availability Statement:** The supporting information on the dataset used in this study is included in the supplementary file.

## Abstract

Access to clean and efficient cooking fuel is crucial for promoting good health, safeguarding the environment, and driving economic growth. Despite efforts to promote the adoption of cleaner alternatives, traditional solid fuels such as charcoal and firewood remain prevalent in Ghana. In this study, we utilized a statistical mechanical model as a framework to explore the statistical relationship between socio-economic factors such as educational attainment, wealth status, place of residence, and cooking fuel choices. We analysed data from the Ghana Malaria Indicator Survey (GMIS) conducted in 2019, involving a total of 2,942 women of reproductive age. The findings revealed that 13.77% of participants preferred using LPG fuels for cooking, while 86.23% preferred non-LPG fuels for their cooking needs. The data indicated that among LPG users, 96.54% are educated women of reproductive age, and 3.46% are non-educated women of reproductive age. Among these, 95.31% are non-poor, and 4.69% are poor. Additionally, 21.73% reside in rural areas, while 78.27% live in urban areas. The data also showed that among non-LPG fuel users, 68.70% are educated women of reproductive age, and 31.30% are non-educated women of reproductive age. Among this group, 16.04% are non-poor, and 83.96% are poor. Furthermore, 67.24% reside in rural areas, and 32.76% live in urban areas. Our findings showed that in the absence of social interaction, a woman's wealth status has a relationship to her choice of fuel for cooking. Additionally, women of reproductive age in rural areas with some education demonstrated a significant private incentive (40.12%) to use LPG, implying a positive correlation between education and the use of LPG for cooking. However, when social interactions are considered, factors such as education, wealth status, and place of residence have significant relationships with a woman's decision about fuel choice. The interaction strength among women of reproductive age in urban areas with some education shows a negative estimate (-4.06%), suggesting that there is no significant imitative effect. The study further suggests that urban women of reproductive age who are poor exert a greater influence on

**Funding:** The author(s) received no specific funding for this work.

**Competing interests:** The author declares that he has no competing interests.

their urban counterparts who are not poor when social interaction is incorporated. Women of reproductive age in rural areas with some form of education exert a greater influence on women of reproductive age in rural areas with no form of education. We recommend that the government of Ghana and its stakeholders focus on leveraging the influence of urban poor women and educated rural women through community-led programs and educational campaigns. Financial support mechanisms like microfinance and subsidies, alongside reliable LPG infrastructure, can make access easier for these target groups. Tailored communication strategies, peer-to-peer learning, and collaboration with local institutions are crucial for spreading awareness and encouraging the adoption of LPG.

## Introduction

The fuel chosen for daily household activities like food preparation and heating is a routine issue faced by women. They often spend a significant amount of time collecting and cooking with traditional fuels, such as biomass (firewood, charcoal, and other crop residues) [1, 2].

Globally, many low- and middle-income countries (LMICs) rely exclusively on solid fuels (firewood and biomass) for domestic needs. It is estimated that around 650 million people in sub-Saharan Africa (SSA) will depend on these fuels in unhealthy ways until 2040 [3]. The World Health Organization (WHO) survey of 18 African countries revealed that over 90% of households use solid fuels for lighting, heating, and cooking [4]. In Ghana, approximately 80% of households rely on traditional fuels [5].

The persistent use of traditional fuels has detrimental health and environmental effects, especially in developing countries like Ghana [6]. This fuel consumption contributes to land degradation, especially in densely populated areas. Charcoal production which is prevalent in the Savannah zone of northern Ghana, exacerbates deforestation [7]. From 2000 to 2015, Ghana experienced an average annual deforestation rate of 0.3% [4]. Burning these fuels also increases greenhouse gas (GHG) emissions [8] and black carbon [9]. A significant portion of the Ghanaian population has been exposed to harmful particulate matter levels, surpassing WHO recommended limits [4].

Using solid fuels daily contributes to both outdoor and indoor air pollution in LMICs [4]. The WHO attributes the deaths of seven million people globally to exposure to these pollutants [4]. Pollution from unclean cooking fuel was linked to nearly four million premature deaths in 2016 which is more than the combined deaths from tuberculosis, malaria, and HIV [4, 10].

Prolonged exposure to air pollution from traditional cooking fuels poses severe health risks, especially for women and sometimes their children [10]. Such exposure during pregnancy is associated with increased risks of low birth weight, stillbirths, neonatal mortality, and pregnancy-induced hypertensive disorders [11–13]. Regular inhalation of particulates can infiltrate maternal bloodstreams, leading to cell damage and inflammation, among other issues [14].

The myriad of adverse effects from the constant use of traditional cooking fuels underscores the urgency of transitioning to cleaner and more sustainable energy sources. The Ghanaian government's efforts to promote cleaner cooking fuels led to the introduction of the Rural Liquefied Petroleum Gas (LPG) Promotion and the Cylinder Recirculation Model programs [15]. Financial incentives for LPG transporters in rural areas and government subsidies were launched to make LPG more accessible [16].

However, despite these efforts, under 25% of Ghanaian households use LPG or electricity [17, 18]. This limited uptake could hinder Ghana's pursuit of the Sustainable Development

Goals (SDGs), particularly, ensuring universal access to clean energy (SDG 7), combating climate change (SDG 13), fostering sustainable communities (SDG 11), and promoting health and well-being (SDG 3) by 2030.

In India, Gould and Urpelainen [19] noted the importance of clean cooking fuels in reducing indoor air pollution and improving public health. They found that LPG adoption correlates strongly with education, but not necessarily with positive attitudes toward clean fuels [19]. Despite the clear health benefits of clean cooking fuels, gender inequality and education gaps hinder full appreciation and adoption [20].

Educational attainment, particularly for women, is influenced by various socio-demographic factors that create complex patterns in decision-making. Understanding these influences has become critical as governments and stakeholders aim to improve educational access and reduce disparities. Recent studies suggest that statistical mechanics models, such as the multipopulation Curie-Weiss model, provide powerful frameworks for examining these patterns by capturing the impact of social interactions on individual choices [21, 22]

This paper focuses on understanding the complex interactions among women within a reference group and how these relationships influence their collective actions [22–24]. Self-organization, a facet of group behaviour, is evident in various contexts such as biology, ecology, and socio-economic systems [21, 25, 26]. In these situations, even minor alterations to the socio-economic landscape can lead to significant changes in the behaviour of a large group. For example, the introduction of a small immigrant group could result in changes in language accents, whereas strategic interventions by authorities might substantially decrease criminal activities [21, 27, 28].

The idea that large-scale behaviours emerge from changes in the interactions among their components has roots in statistical mechanics. Spin models, especially those explaining ferromagnetism, have demonstrated the phenomenon of phase transitions, as highlighted in [29, 30]. The mean-field Ising model, introduced in [29], stands as a prime example of these models. Its application extends across diverse fields such as social studies, finance, chemistry, and ecology [31–34]. The variant of the multi-population mean-field Ising model, has gained popularity, particularly in studying magnetism in anisotropic substances, as described in [35]. Additionally, scholars in mathematical physics, including Contucci, Gallo, and Barra, have utilized this model in their research, as cited in [36, 37]. The significance of integrating statistical mechanical models into social science research is underscored by insights from [22–24]. This integration offers a valuable avenue for scholars to further explore the profound effects of social interactions on societal dynamics, as suggested in [21].

The Curie-Weiss model for multiple populations provides an analytical framework to study binary decisions. This includes choices such as whether to pursue further education or enter the workforce. These decisions are influenced by the individual's socio-economic environment, leading to the observation that individuals with similar socio-economic backgrounds tend to exhibit comparable decision-making patterns. In contrast, those from different backgrounds are likely to make different choices. This premise has been supported by prior research, as detailed in studies by [37]. Further exploration has been conducted into a multi-group version of this model, as seen in works by [36, 37]. These studies extended the use of the modified Curie-Weiss approach to multiple demographic groups, ensuring a constant proportion within each group, regardless of overall population size. For example, [37] applied this framework to analyse factors such as suicidal tendencies and marriage trends in Italy, with a specific focus on living location as the socio-economic factor. Additionally, research by [21] has examined the impact of variables such as gender and domicile on educational outcomes across five emerging nations in Africa.

Recent research across health, education, and financial security in Ghana highlights the instrumental role of the Curie-Weiss model in understanding how socio-economic and health factors shape decision-making among women. For example, one study applies this model to analyse the relationship between anaemia, pregnancy status, and the use of insecticide-treated bed nets. The findings indicate that anaemic women, particularly those who are pregnant, are more likely to use these nets, emphasizing the crucial role of malaria prevention in reducing anaemia within high-risk populations [38]. The Curie-Weiss model's application extends to educational attainment, where researchers examine the influence of marital status, poverty, and rural residency on Ghanaian women's education. The model reveals that social interactions and socio-demographic factors create substantial barriers to educational progress, particularly for women from low-income and rural backgrounds. This study suggests that targeted policies focused on poverty alleviation, infrastructure improvements, and gender-sensitive approaches could enhance educational outcomes [39]. In financial decision-making, the Curie-Weiss model is also essential for exploring factors that influence motor insurance choices. The study indicates that gender and residence type affect preferences between comprehensive and third-party insurance, with rural residents and women showing a preference for comprehensive coverage. This pattern likely reflects a perceived need for greater security within these demographic groups. The study advocates for improved insurance education to help rural drivers make informed policy decisions [40].

For a deeper exploration of the potential applications of statistical mechanical models in social sciences and energy domains, we suggest consulting the following references: [22–24, 41]. While prior findings have studied factors affecting the choice of cooking fuels in Ghana, few have focused on women of reproductive age [42, 43]. Our unique contribution is to identify a statistical relationship among educational attainment, wealth status, and place of residence in relation to cooking fuel choices. To achieve this, we employ the multipopulation Curie-Weiss model to study the cooking fuel preferences of Ghanaian women of reproductive age.

## Methods

### Study design and data

The study introduces an analytical cross-sectional analysis focusing on females throughout Ghana's original ten regions. This research utilized the 2019 Ghana Malaria Indicator Survey (GMIS) data, which is grounded in the 2010 Population and Housing Census (PHC) conducted by the Ghana Statistical Service (GSS). It's noteworthy that Ghana experienced a regional restructuring in 2019, expanding from 10 to 16 regions, adding 260 more districts and municipalities. However, the 2019 GMIS only covered the original ten regions as delineated in the 2010 PHC due to the new regions' recent establishment. The purpose of the GMIS is to gather detailed information regarding malaria and its effects, providing essential insights for both the national and regional levels, specifically for the ten regions established in the 2010 PHC: Western, Central, Greater Accra, Volta, Eastern, Ashanti, Brong Ahafo, Northern, Upper East, and Upper West. The survey utilized the census enumeration areas (EAs) from the 2010 PHC, as provided by the GSS [44].

The 2019 GMIS study utilized a two-stage sampling process, segmented into 20 distinct groups based on the urban-rural classification in each region. Initially, 200 Enumeration Areas (EAs) were selected, split into 97 urban and 103 rural EAs, following a method of independent choice and size consideration in each group, as elaborated in [44]. In the subsequent stage, 30 households were chosen from each group, cumulatively amounting to 6,000 households. The selection framework within each group was arranged for implicit stratification and

proportionate distribution at more granular administrative levels. For an in-depth understanding of the 2019 GMIS sampling, inclusive of data gathering techniques, instruments, and quality assurance, the full report is available in [44]. This study's analysis was particularly focused on the data from 2,942 women of reproductive age, aged between 15 and 49 years.

## Study variables

The study examined the cooking fuel preferences of women of reproductive age, using these preferences as the dependent variable. Cooking fuel choices were categorized into two groups: the use of LPG as a cooking fuel (coded as +1) and the use of non-LPG options such as wood, straw, shrubs, grass, and charcoal (coded as -1). The study also considered several independent variables: type of residence (rural or urban), educational attainment (with or without formal education), and the wealth status (poor or non-poor) of the households where these women were surveyed.

## Ethics statement

The ethical protocols for the 2019 Ghana Maternal Health Survey (GMIS) received approval from the Ghana Health Service Ethical Review Committee and the ICF International Institutional Review Board. Participation in this survey was entirely voluntary, with participants being fully informed about the potential risks and benefits involved. Only those who consented to participate by signing an informed consent form were included in the study. There was no specific ethics ID or approval number assigned. Participants were required to give their informed consent before they could take part in the study.

## Statistical analysis

### The Curie-Weiss Hamiltonian

Let $N$ denote the population size of women of reproductive age, and we represent each individual's decision with a binary action:

$$\eta_a = \begin{cases} +1, & \text{if individual } a \text{ uses LPG as a cooking fuel,} \\ -1, & \text{if individual } a \text{ uses non-LPG as a cooking fuel} \end{cases}$$

for $a \in \{1, \cdots, N\}$. We now define $H_N$ for any $\eta \in \Omega_N = \{-1, +1\}^N$ as

$$H_N(\eta) = \frac{1}{2N} \sum_{a,b=1}^{N} \bar{J}_{ab} \eta_a \eta_b + \sum_{a=1}^{N} h_a \eta_a. \tag{0.1}$$

The notation $\eta = (\eta_1, \eta_2, \ldots, \eta_N) \in \Omega_N$ can be used to depict the choices made by all $N$ individuals. $H_N(\eta)$ returns the level of satisfaction or utility of the entire population resulting from their choices represented by $\eta$. Higher values of $H_N(\eta)$ indicate greater satisfaction or utility within the population. The Hamiltonian $H_N(\eta)$ consists of two components: the first part models the social incentives of individuals in the population, while the second part models the private incentives of each individual [45]. In this context, $\eta_a$ represents the choice made by an individual $a$, $\bar{J}_{ab}$ measures the influence that the individual $a$ has on individual $b$ which represents the strength of social incentive, and $h_a$ represents the external influence. A positive value of $\bar{J}_{ab}$ indicates that conformity or imitation is rewarded, whereas a negative value suggests that imitation or conformity is not rewarded [21].

## Multipopulation Curie-Weiss model

Our main premise is that women of reproductive age with similar characteristics exhibit similar behaviour, whereas women of reproductive age with diverse attributes display diverse behaviours. This premise plays a crucial role in redefining the parameters of the Curie-Weiss Hamiltonian as shown in Eq (0.1). Consequently, our primary objective is to determine a suitable parameterization for the interaction coefficient $\bar{J}_{ab}$ and develop a systematic approach for estimating the model's parameters using data. In line with our discrete choice model, each individual $a$ is assigned a set of $k$ characteristics:

$$T_a = \left( T_a^{(1)}, T_a^{(2)}, .... T_a^{(k)} \right) \tag{0.2}$$

where the $T_a^{(j)} \in \{0, 1\}^k$ for $j = 1, 2, \ldots, k$. We consider the scenario in which the socio-economic attributes under examination are educational attainment represented by $T_a^{(1)}$, wealth status represented by $T_a^{(2)}$, and place of residence represented by $T_a^{(3)}$ with:

$$T_a^{(1)} = \begin{cases} 1, & \text{if woman } a \text{ has some form of education} \\ 0, & \text{if woman } a \text{ has no education} \end{cases} \tag{0.3}$$

$$T_a^{(2)} = \begin{cases} 1, & \text{if woman } a \text{ is non poor} \\ 0, & \text{if woman } a \text{ is poor} \end{cases} \tag{0.4}$$

and

$$T_a^{(3)} = \begin{cases} 1, & \text{if woman } a \text{ dwells in rural} \\ 0, & \text{if woman } a \text{ dwells in urban.} \end{cases} \tag{0.5}$$

If the value of an attribute is zero for an individual, then it means that a particular attribute does not contribute to the private incentive of that individual. Because $h_a$ is what motivates an individual to make a choice or decision, it is reasonable for it to become dependent on the vector of socio-economic attributes $T_a$ since every rational person considers her status before making a choice or decision. Therefore, it can be written as a function of $T_a$ as follows: let $\beta_j$ for $j = 0, \cdots, k$ be the component of the vector $\beta = (\beta_0, \beta_1, \cdots, \beta_k)$ and assume $\beta_j$ does not depend on the specific individual $a$. The vector $\beta$ tells us the relative weight or importance that the various socio-economic attributes has or measures the private incentive for each attribute when an individual is making a decision. In our case this leads to

$$h_a = \sum_{j=1}^{k} \beta_j T_a^{(j)} + \beta_0. \tag{0.6}$$

Here, $\beta_0$ represents the fundamental private incentive that everyone person possesses, irrespective of their unique socio-economic characteristics [21].

## Estimation

The least squares method is used to estimate the model parameters. As a result, we must identify the parameter settings that minimize

$$\sum_g \left[ \bar{m}_g - \tanh(U_g) \right]^2 \tag{0.7}$$

where $\bar{m}_g$ is the average choice of group $g$. Because $\tanh(U_g)$ is non-linear, the computation will take an extremely long time, see [21]. In the interaction scenario, the independent variables are correlated. As a result, the least squares method is rendered ineffective. In that case, the partial least squares estimation method will be utilized. In the following section, we utilized the previously described estimation method to assess the parameters concerning the fuel choices of Ghanaian women of reproductive age. We would then estimate these parameters based on their socio-economic characteristics, including educational attainment, wealth status, and place of residence. We utilized MATLAB software version R2016a for carrying out our analyses and obtaining results.

In this research, we employed a statistical mechanical framework to investigate the relationships among socio-economic variables such as education level, wealth status, and living environment, and their impact on the choice of cooking fuels. We analysed data from the 2019 Ghana Malaria Indicator Survey, which included responses from 2,942 women of reproductive age. The data revealed that 13.77% of participants preferred using LPG fuels for cooking, while 86.23% preferred non-LPG fuels for their cooking needs. The data indicated that among LPG users, 96.54% are educated women of reproductive age, and 3.46% are non-educated women of reproductive age. Among these, 95.31% are non-poor, and 4.69% are poor. Additionally, 21.73% reside in rural areas, while 78.27% live in urban areas. The data also showed that among non-LPG fuel users, 68.70% are educated women of reproductive age, and 31.30% are non-educated women of reproductive age. Among this group, 16.04% are non-poor, and 83.96% are poor. Furthermore, 67.24% reside in rural areas, and 32.76% live in urban areas.

Here we are looking at women of reproductive age that have been partitioned according to three attributes $T_a^{(1)}$, $T_a^{(2)}$ and $T_a^{(3)}$ representing educational attainment, wealth status, and place of residence. Table 1 shows the partition of the population into eight subsection, that is, women of reproductive age with some form of education and in rural, women of reproductive age with no education and in rural, women of reproductive age with some form of education and in urban, women of reproductive age with no education and in urban, women of reproductive age of reproductive age in rural and non-poor, women of reproductive age in rural and poor, women of reproductive age in urban and non-poor and women of reproductive age in urban and poor.

Tables 2 and 3 below show the partitioning of the population on the use of LPG and non-LPG with respect to their attributes.

We got eight groups that were indexed with this type of partitioning $a = 1, 2, \cdots, 8$. In particular,

$a = 1$ represents the group of women of reproductive age with some form of education and in rural,

$a = 2$ represents the group women of reproductive age with no education and in rural,

$a = 3$ represents the group of women of reproductive age with some form of education and in urban,

$a = 4$ represents the group of women of reproductive age with no education and in urban,

**Table 1. Population classification based on attributes.**

| Attributes | Educational Attainment | | Wealth Status | |
|---|---|---|---|---|
| Residence | Some form of education | No education | Non-poor | Poor |
| Rural | 1179 | 615 | 175 | 1619 |
| Urban | 955 | 193 | 618 | 530 |

**Table 2. LPG users.**

| Attributes | Educational Attainment | | Wealth Status | |
|---|---|---|---|---|
| Residence | Some form of education | No education | Non-poor | Poor |
| Rural | 86 | 2 | 76 | 12 |
| Urban | 305 | 12 | 310 | 7 |

**Table 3. Non-LPG (wood, straw/shrubs/grass, and charcoal) users.**

| Attributes | Educational Attainment | | Wealth Status | |
|---|---|---|---|---|
| Residence | Some form of education | No education | Non-poor | Poor |
| Rural | 1093 | 613 | 99 | 1607 |
| Urban | 650 | 181 | 308 | 523 |

**Table 4. Attributes of the population.**

| | ATTRIBUTES | | | | | |
|---|---|---|---|---|---|---|
| Cases | Educational Attainment($T_a^{(1)}$) | | Wealth Status($T_a^{(2)}$) | | Residence($T_a^{(3)}$) | |
| | No education | Some form of education | Poor | Non-Poor | Urban | Rural |
| 1 | 0 | 1 | 1 | 0 | 1 | 0 |
| 2 | 1 | 0 | 0 | 1 | 1 | 0 |
| 3 | 0 | 1 | 0 | 1 | 0 | 1 |
| 4 | 1 | 0 | 1 | 0 | 0 | 1 |
| 5 | 0 | 1 | 1 | 0 | 1 | 0 |
| 6 | 1 | 0 | 0 | 1 | 1 | 0 |
| 7 | 0 | 1 | 0 | 1 | 0 | 1 |
| 8 | 1 | 0 | 1 | 0 | 0 | 1 |

$a$ = 5 represents the group of women of reproductive age in rural and non-poor,

$a$ = 6 represents the group of women of reproductive age in rural and poor,

$a$ = 7 represents the group of women of reproductive age in urban and non-poor,

$a$ = 8 represents the group of women of reproductive age in urban and poor.

We then computed the values of the attributes $T_a^{(1)}$, $T_a^{(2)}$ and $T_a^{(3)}$ representing educational level, wealth status, and place of residence depending on the group a woman of reproductive age belongs, for the eight cases generated in Table 4 below. The values assigned to the attributes in each of the eight cases describe the relevance of that attribute to the private incentive part of the Hamiltonian. For instance in Case 1, being a woman of reproductive age with no form of education as an attribute does not contribute to the private incentive of a group, while being a woman of reproductive age with some form of education contributes to the private incentive of a group.

## Results

### Non interacting case

The group's decision to use LPG or non-LPG as cooking fuel depends on whether the sum of the individual private incentives, $\beta_j$, is positive or negative. A positive sum will lead to the

**Table 5. Estimates for the non-interacting model.**

| Cases | Estimate |
|---|---|
| Case 1: women of reproductive age with some form of education and in rural | $\beta_1 = 0.4012$ |
| | $\beta_2 = 0.1324$ |
| | $\beta_3 = -0.0523$ |
| | $\beta_0 = -0.4891$ |
| Case 2: women of reproductive age with no education and in rural | $\beta_1 = -0.0815$ |
| | $\beta_2 = 0.1394$ |
| | $\beta_3 = -0.1280$ |
| | $\beta_0 = -0.1568$ |
| Case 3: women of reproductive age with some form of education and in urban | $\beta_1 = 0.0575$ |
| | $\beta_2 = 0.1310$ |
| | $\beta_3 = 0.2914$ |
| | $\beta_0 = -0.4317$ |
| Case 4: women of reproductive age with no education and in urban | $\beta_1 = -0.3371$ |
| | $\beta_2 = 0.0985$ |
| | $\beta_3 = -0.1103$ |
| | $\beta_0 = -0.0457$ |
| Case 5: women of reproductive age in rural and non-poor | $\beta_1 = 0.4012$ |
| | $\beta_2 = 0.1324$ |
| | $\beta_3 = -0.0523$ |
| | $\beta_0 = -0.4891$ |
| Case 6: women of reproductive age in rural and poor | $\beta_1 = -0.0815$ |
| | $\beta_2 = 0.1394$ |
| | $\beta_3 = -0.1280$ |
| | $\beta_0 = -0.1568$ |
| Case 7: women of reproductive age in urban and non-poor | $\beta_1 = 0.0575$ |
| | $\beta_2 = 0.1310$ |
| | $\beta_3 = 0.2914$ |
| | $\beta_0 = -0.4317$ |
| Case 8: women of reproductive age in urban and poor | $\beta_1 = -0.3371$ |
| | $\beta_2 = 0.0985$ |
| | $\beta_3 = -0.1103$ |
| | $\beta_0 = -0.0457$ |

choice of LPG, while a negative sum will result in the selection of non-LPG. The calculated estimates for $\beta_j$ are shown in Table 5. The sums of $\beta_j$ for cases 1, 2, 4, 5, 6, and 8 are negative, indicating that individuals in these cases are more likely to choose non-LPG as their cooking fuel when there is no social interaction. Conversely, the positive sums of $\beta_j$ in cases 3 and 7 suggest that women of reproductive age are inclined to opt for LPG as their cooking fuel in the absence of social interaction.

If social interaction is absent, the estimates suggest that wealth status ($\beta_2$) is related to the choice of cooking fuel, as the private incentive for wealth status is positive in Cases 1 through 8. The negative value of the base private incentive ($\beta_0$) in all eight instances indicates that individuals would opt for some form of non-LPG cooking fuel option if there is no social interaction present.

### Interacting case

In the interactive model, the utility function considers both social and private incentives. If the social incentive, $\bar{J}_{ab}$, is positive, individuals within the groups tend to mimic each other's choices. Conversely, when $\bar{J}_{ab}$ takes negative values, women of reproductive age in those groups are likely to make dissimilar choices instead of adhering to the norm. The decision to use LPG or non-LPG as cooking fuel depends on the sum of the private incentives, represented by $\beta_j$. If this sum is positive, the group will opt for LPG; if negative, they will choose non-LPG.

Note that $\bar{J}_{11}$ represents the strength of interaction among women of reproductive age, with some form of education and living in rural areas, interacting with each other. As shown in Table 6, the estimated value of $\bar{J}_{11}$ is positive, indicating that imitation is considered a reward in this group. On the other hand, $\bar{J}_{33}$ refers to the strength of interaction among women of reproductive age with some form of education, living in urban areas, interacting with each other. The estimated value of $\bar{J}_{33}$ is negative, suggesting that imitation is not seen as a reward in this group. $\bar{J}_{61}$ has the highest estimate for social interaction, indicating that women of reproductive age in rural areas, who are poor, have a greater influence on their educated peers also living in rural areas.

Table 5 shows that educational attainment, wealth status, and place of residence have a relationship with the choice of cooking fuel when social interaction is present, as indicated by the positive values of $\beta_1$, $\beta_2$, and $\beta_3$. Additionally, the value of $\beta_0$ is zero, which implies that in situations where there is social interaction, women of reproductive age consider their education, wealth status, or residence when deciding whether to use LPG as a cooking fuel.

## Discussions

This study provides valuable insights into the influence of socio-economic factors on cooking fuel choices among Ghanaian women of reproductive age, using statistical mechanics models to explore both individual preferences and social interactions. Our analysis, drawing from the Ghana Malaria Indicator Survey (GMIS) data, identifies that educational attainment, wealth status, and place of residence are significant determinants in the decision to adopt liquefied petroleum gas (LPG) as a cleaner alternative fuel. The study highlights a clear positive relationship between education level and LPG adoption. Women with some formal education, particularly those residing in urban areas, displayed a higher likelihood of choosing LPG over traditional fuels. This aligns with global trends where education fosters awareness of the health and environmental benefits of clean energy sources. Our data indicated that among the LPG users 96.54% are educated women of reproductive age and 3.46% are non-educated women of reproductive age, indicating that educated women may recognize the long-term health advantages of cleaner fuels, reinforcing previous studies from other developing countries, where education positively correlates with LPG adoption [46–48].

The study further identifies wealth status as a significant predictor of fuel choice, with wealthier women more inclined to use LPG. In Ghana, the initial cost and recurring expenses associated with LPG can be prohibitive for lower-income households, a trend observed in other low- and middle-income countries as well [46, 49]. The positive relationship between wealth and LPG adoption found in this study supports the importance of economic support programs such as subsidies and microfinance to make LPG more accessible to poorer households. This finding mirrors existing literature emphasizing the need to address economic barriers in promoting clean cooking solutions in developing countries [50, 51]. The role of social interactions provides a significant perspective on fuel choice behaviours. Additionally, poor women appeared to exert a notable influence on wealthier peers in urban settings and also,

**Table 6. Estimates for the interacting model.**

| Parameter | Estimate |
|---|---|
| $\bar{J}_{11}$ | 0.1203 |
| $\bar{J}_{12}$ | 0.0730 |
| $\bar{J}_{13}$ | 0.0412 |
| $\bar{J}_{14}$ | 0.0202 |
| $\bar{J}_{15}$ | 0.0027 |
| $\bar{J}_{16}$ | 0.1905 |
| $\bar{J}_{17}$ | -0.0002 |
| $\bar{J}_{18}$ | 0.0616 |
| $\bar{J}_{21}$ | 0.0996 |
| $\bar{J}_{22}$ | 0.0604 |
| $\bar{J}_{23}$ | 0.0341 |
| $\bar{J}_{24}$ | 0.0167 |
| $\bar{J}_{25}$ | 0.0023 |
| $\bar{J}_{26}$ | 0.1578 |
| $\bar{J}_{27}$ | -0.0002 |
| $\bar{J}_{28}$ | 0.0510 |
| $\bar{J}_{31}$ | -0.1184 |
| $\bar{J}_{32}$ | -0.0718 |
| $\bar{J}_{33}$ | -0.0406 |
| $\bar{J}_{34}$ | -0.0199 |
| $\bar{J}_{35}$ | -0.0027 |
| $\bar{J}_{36}$ | -0.1875 |
| $\bar{J}_{37}$ | 0.0002 |
| $\bar{J}_{38}$ | -0.0607 |
| $\bar{J}_{41}$ | -0.0620 |
| $\bar{J}_{42}$ | -0.0376 |
| $\bar{J}_{43}$ | -0.0213 |
| $\bar{J}_{44}$ | -0.0104 |
| $\bar{J}_{45}$ | -0.0014 |
| $\bar{J}_{46}$ | -0.0982 |
| $\bar{J}_{47}$ | 0.0001 |
| $\bar{J}_{48}$ | -0.0318 |
| $\bar{J}_{51}$ | -0.0072 |
| $\bar{J}_{52}$ | -0.0044 |
| $\bar{J}_{53}$ | -0.0025 |
| $\bar{J}_{54}$ | -0.0012 |
| $\bar{J}_{55}$ | -0.0002 |
| $\bar{J}_{56}$ | -0.0114 |
| $\bar{J}_{57}$ | 0.0000 |
| $\bar{J}_{58}$ | -0.0037 |
| $\bar{J}_{61}$ | 0.2118 |
| $\bar{J}_{62}$ | 0.1285 |
| $\bar{J}_{63}$ | 0.0726 |
| $\bar{J}_{64}$ | 0.0355 |
| $\bar{J}_{65}$ | 0.0048 |

(*Continued*)

**Table 6.** (Continued)

| Parameter | Estimate |
|---|---|
| $\bar{J}_{66}$ | 0.3354 |
| $\bar{J}_{67}$ | -0.0004 |
| $\bar{J}_{68}$ | 0.1085 |
| $\bar{J}_{71}$ | -0.1329 |
| $\bar{J}_{72}$ | -0.0806 |
| $\bar{J}_{73}$ | -0.0455 |
| $\bar{J}_{74}$ | -0.0223 |
| $\bar{J}_{75}$ | -0.0030 |
| $\bar{J}_{76}$ | -0.2105 |
| $\bar{J}_{77}$ | 0.0003 |
| $\bar{J}_{78}$ | -0.0681 |
| $\bar{J}_{81}$ | -0.1111 |
| $\bar{J}_{82}$ | -0.0674 |
| $\bar{J}_{83}$ | -0.0381 |
| $\bar{J}_{84}$ | -0.0187 |
| $\bar{J}_{85}$ | -0.0025 |
| $\bar{J}_{86}$ | -0.1760 |
| $\bar{J}_{87}$ | 0.1002 |
| $\bar{J}_{88}$ | -0.1570 |
| $\beta_1$ | 1.311 |
| $\beta_2$ | 0.9678 |
| $\beta_3$ | 0.7887 |
| $\beta_0$ | 0 |

rural poor women of reproductive age exerted a notable influence on urban poor women of reproductive age, hinting at the potential of community-driven initiatives to drive change. For instance, poorly educated women in rural communities tend to conform to the prevalent norms in their social circles, which may inhibit the transition to cleaner fuels. Conversely, urban poor women appear to have an unexpectedly strong influence on wealthier peers, suggesting that communal bonds and shared challenges in accessing resources play a significant role in LPG adoption. The analysis, using the Curie-Weiss multipopulation model, reveals that while urban women show a limited imitative effect, women in rural areas exhibit a notable tendency to mimic the fuel choices of their social counterparts. This suggests that tailored interventions encouraging LPG adoption should consider these local dynamics and the use of influential community members to promote behavioural change.

Addressing the study's limitations, we acknowledge that our reliance on cross-sectional data from the Ghana Multiple Indicator Survey (GMIS) limits our ability to infer causal relationships. Future research could benefit from longitudinal studies that explore how changes in socio-economic status over time affect cooking fuel choices. Given the clear relationship between socio-economic factors, such as educational attainment, wealth status, place of residence and cooking fuel choices, policies aimed at promoting cleaner cooking fuels in Ghana must consider economic incentives, provide free education for women of reproductive age, and implement community-based interventions tailored to the specific needs of rural and urban populations.

**Table 7. Variance of $\bar{m}_g$ and $U_g$ explained by the latent vectors and RMSEP.**

| Latent Vectors | Percentage of Explained Variances for $\bar{m}_g$ | Cumulative Percentage of Explained Variances for $\bar{m}_g$ | Percentage of Explained Variances for $U_g$ | Cumulative Percentage of Explained Variances for $U_g$ | RMSEP for $\bar{m}_g$ | RMSEP for $U_g$ |
|---|---|---|---|---|---|---|
| 1 | 27.47 | 27.47 | 39.84 | 39.84 | 1.0761 | 0.1918 |
| 2 | 10.89 | 38.36 | 15.98 | 55.82 | 0.8509 | 0.1054 |
| 3 | 9.84 | 48.2 | 7.84 | 63.66 | 0.7349 | 0.0548 |
| 4 | 10.54 | 58.74 | 0.34 | 64.00 | 0.6528 | 0.01 |
| 5 | 5.42 | 64.16 | 0 | 64.00 | 0.4339 | 0 |
| 6 | 14.78 | 78.94 | 16.11 | 80.11 | 0.3543 | 0 |
| 7 | 15.76 | 94.70 | 17.20 | 97.31 | 0.5282 | 0.07681 |

## Model diagnostics and validation

In this section, we will assess the goodness of fit of our interacting model to the data. To estimate the parameters of the interacting model proposed in [52], we employed the partial least squares (PLS) method. Our dependent variable is represented by $U_g$, while our independent variable is $\bar{m}_g$. PLS is a technique used to predict a set of dependent variables based on a set of independent variables, which results in orthogonal factors or latent vectors derived from the independent variables. These latent vectors are then sorted by decreasing order of their eigenvalues, and the number of latent vectors used is determined by those that best explain the covariance between the independent and dependent variables [53].

Table 7 displays the variance explained by the latent vectors used in estimating the independent variables ($\bar{m}_g$) and the dependent variable ($U_g$), as well as their root mean square error of prediction (RMSEP). According to Table 7, the first seven latent vectors explain 97.31% of the variance in $U_g$ and 94.70% of the variance in $\bar{m}_g$. The variances explained by both the dependent and independent variables are sufficient for modelling or prediction. The RMSEP decreases as the number of latent vectors increases until it reaches the sixth latent vector, at this point it increases again for both $\bar{m}_g$ and $U_g$.

## Conclusion

This study highlights the complex socio-economic and social factors influencing cooking fuel choices among Ghanaian women of reproductive age, revealing patterns that go beyond individual decision-making and extend into the dynamics of communities and social groups. By examining education, wealth status, and place of residence through the lens of statistical mechanics, we provide a fresh perspective on the motivations and barriers that shape these choices.

Our analysis shows that educated women, particularly in urban areas, are more inclined to adopt LPG, supporting the role of educational attainment in environmental health and sustainable energy choices. This trend emphasizes that education does not just serve as a pathway to individual empowerment but also significantly influences lifestyle and health-related decisions, benefiting both households and the broader community. On the other hand, wealth constraints among poorer households emerged as a key barrier, limiting access to LPG, which is often seen as a cleaner but relatively expensive alternative. This is particularly pertinent in rural settings, where economic limitations compound logistical challenges, making clean fuel options both less accessible and less adopted.

Social interactions also play a powerful, albeit context-dependent, role. In rural areas, for example, poor, uneducated women hold substantial influence over their peers, indicating that

social norms and community dynamics may often outweigh individual preferences in these contexts. This social influence suggests an opportunity for community-based initiatives that could drive significant shifts in fuel choices through targeted interventions. By recognizing the varying roles of socio-economic and social factors across different communities, this study provides a detailed understanding of cooking fuel choices, adding a valuable dimension to existing literature on energy behaviour and adoption in developing countries.

Programs focused on raising awareness about the health benefits of clean fuels, especially among less educated women, could catalyse change. Furthermore, economic policies like subsidies and financial support mechanisms would alleviate cost barriers for poorer households. Additionally, improving LPG infrastructure, especially in rural areas, would enhance accessibility and potentially accelerate adoption. In essence, promoting LPG adoption is not simply a matter of technology or affordability; it requires a deeper engagement with the social fabric and economic realities of each community. Understanding these dynamics not only aids in crafting more effective policies but also aligns with broader goals, such as reducing health risks from traditional fuel use and advancing sustainable development objectives.

## Supporting information

**S1 Dataset. Supporting information on the dataset.**
(XLSX)

**S1 File. Supplementary information on the methodology.**
(PDF)

## Author Contributions

**Conceptualization:** Richard Kwame Ansah, Richard Kena Boadi, William Obeng-Denteh, Kassim Tawiah.

**Data curation:** Richard Kwame Ansah, Killian Asampana Asosega.

**Formal analysis:** Richard Kwame Ansah, Killian Asampana Asosega, Kassim Tawiah.

**Investigation:** Killian Asampana Asosega, Kassim Tawiah.

**Methodology:** Richard Kwame Ansah, Richard Kena Boadi, William Obeng-Denteh.

**Resources:** Kassim Tawiah.

**Software:** Richard Kwame Ansah, William Obeng-Denteh, Killian Asampana Asosega.

**Supervision:** Richard Kwame Ansah, Richard Kena Boadi, William Obeng-Denteh.

**Writing – original draft:** Richard Kwame Ansah.

**Writing – review & editing:** Richard Kwame Ansah.

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
