## [Decision Letter · Decision Letter 0]

14 Oct 2024

PONE-D-24-15863Exploring Cooking Fuel Choices among Ghanaian Women of Reproductive Age: A Socio-Economic Analysis from a Statistical Mechanics PerspectivePLOS ONE

Dear Dr. Ansah,

Thank you for submitting your manuscript to PLOS ONE. After careful consideration, we feel that it has merit but does not fully meet PLOS ONE’s publication criteria as it currently stands. Therefore, we invite you to submit a revised version of the manuscript that addresses the points raised during the process. Please submit your revised manuscript by Nov 28 2024 11:59PM. If you will need more time than this to complete your revisions, please reply to this message or contact the journal office at plosone@plos.org. Please include the following items when submitting your revised manuscript:A rebuttal letter that responds to each point raised by the academic editor and reviewer(s). You should upload this letter as a separate file labeled 'Response to Reviewers'.A marked-up copy of your manuscript that highlights changes made to the original version. You should upload this as a separate file labeled 'Revised Manuscript with Track Changes'.An unmarked version of your revised paper without tracked changes. You should upload this as a separate file labeled 'Manuscript'.

We look forward to receiving your revised manuscript.

Kind regards,

Muhammad Ali, PhD

Academic Editor

PLOS ONE

Journal Requirements:

2. Please note that PLOS ONE has specific guidelines on code sharing for submissions in which author-generated code underpins the findings in the manuscript. In these cases, we expect all author-generated code to be made available without restrictions upon publication of the work. 

Please review our guidelines at https://journals.plos.org/plosone/s/materials-and-software-sharing#loc-sharing-code and ensure that your code is shared in a way that follows best practice and facilitates reproducibility and reuse.

**Additional Editor Comments:**

Dear authors

Thank you for submitting your paper to Plos One. We've now received two reviews for your paper. Both are suggesting major revisions, hence I'm also requesting you to revise your paper according to the reviews then I'll request the reviewers to assess whether they are satisfied with their response.

Thank you

Reviewers' comments:

Reviewer's Responses to Questions

**Comments to the Author**

1. Is the manuscript technically sound, and do the data support the conclusions?

Reviewer #1: Partly

Reviewer #2: Yes

2. Has the statistical analysis been performed appropriately and rigorously? 

Reviewer #1: No

Reviewer #2: No

3. Have the authors made all data underlying the findings in their manuscript fully available?

Reviewer #1: No

Reviewer #2: No

4. Is the manuscript presented in an intelligible fashion and written in standard English?

Reviewer #1: Yes

Reviewer #2: No

5. Review Comments to the Author

Reviewer #1: The paper is on interesting topic and methodology also appears to be ok. However, upon closer reading, some issues arise that need clarification.

1) Authors claim influence of one group/individual on others, however, it is unclear how this "infuence" is measured. Data at hand does not appear to have inherent interactions among population members. Are authors usiing "interaction terms" (multiplicative terms) to measure influence? It is unclear.

2) Despite having many equations in the paper, the most important part is unexplained. How did the authors calculate coefficients? In Table 4, authors disaggregated the population according to their characteristics. In Table 5, authors appear to calculate coefficients for each of these groups. The problem is that, if you define the groups the way they did, there will be no variation in the indepdent variables. Eg: In case 1, everyone will have some education, poor and living in urban areas. This will cause perfect multicollinearity and we'll be unable to estimate coefficients. How did the authors come up with the estimates given in Table 5?

3) where are the estimates in Table 6 coming from? Do authors apply some sort of disaggregation in this estimation as well? it will be important to know how many observations are in each group

4) In Table 2, No Education and Poor have negligible number of observations. Poor people are 5% of the rich ones and uneducated ones are 4% of the educated population. Can we get anything meaninful with such a huge difference?

Reviewer #2: Overall Evaluation: This study explores the relationship between socio-economic factors and cooking fuel choices in Ghana, using data from the 2019 Ghana Malaria Indicator Survey (GMIS). The study highlights that this relationship becomes evident when social interactions are considered. However, several concerns arise. Although the topic is interesting and fills a gap in the literature on fuel use choices, the paper is written in overly technical language, making it difficult for readers to clearly understand the key findings, particularly regarding fuel choices among Ghanaian women of reproductive age. Simplifying the explanations would greatly improve accessibility and comprehension. Some sections, particularly in the discussions and conclusion, could benefit from further elaboration on policy implications or comparison with other studies in similar contexts outside Ghana. I have left my other comments in the attached pdf. The manuscript can be accepted for publications with some major changes.

Abstract: I suggest that these findings may be reported from LPG preference perspective first and then it's division of rural/urban and wealthier classification. To me it should first read 2.92% of educated women of reproductive age prefer LPG. Those who preferred non-LPM are 33.37 %. How about the remaining 59.93% of educated women of reproductive age? They didn't reveal their choice.

Methods: The methodology adopted does provide insights on the preferences of rural education women of reproductive age towards fuel choices, however the methods do not offer a causal interpretation of what drives them to these choices. I understand that the authors acknowledge this causality limitation in this study. With the current constraint, it thinks the authors can explore alternative models such as finding a good instrumental variable in explaining the education and fuel choice relationship using the same cross-sectional data. The paper lacks reference to seminal works in this area that follow the Curie-Weiss Model, and the methods adopted in this study. It is suggested that in the method section be enriched with recent works on similar area using this model.

Results:

The tables are not presented in the style of publication quality tables. A lot can be done with these tables to present in a more scientific way. Please follow empirical papers published in this journal and other journals.

Discussion: The idea that women of reproductive age in urban areas with some form of education show a significant preference for LPG does not bring the novelty in this area. It is almost a collective wisdom that education increase individuals’ awareness about environment and health. It would have been interesting to explore these women’s tendencies towards health (utilizing social interaction from health point of view).

6. PLOS authors have the option to publish the peer review history of their article (what does this mean?). If published, this will include your full peer review and any attached files.

Reviewer #1: No

Reviewer #2: No

---

## [Author Response · Author response to Decision Letter 0]

14 Nov 2024

Journal Requirements: 

Response: Thank you for the reminder. I have carefully reviewed PLOS ONE's style requirements, including the guidelines for file naming, and ensured that my revised manuscript meets all necessary standards.

2. Please note that PLOS ONE has specific guidelines on code sharing for submissions in which author-generated code underpins the findings in the manuscript. In these cases, we expect all author-generated code to be made available without restrictions upon publication of the work. 

Response: Thank you for the information regarding code sharing. I confirm that I will make all author-generated code available without restrictions, in line with PLOS ONE's guidelines, upon publication of the manuscript.

Response: Thank you for the guidance. I will update my submission using the PLOS LaTeX template as requested. I have ensured that the revised document adheres to the submission requirements.

Response: I have moved the ethics statement exclusively to the Methods section and removed it from all other sections. Kindly see line 191 of the revised manuscript.

Reviewer 1 Comments 

1. Authors claim influence of one group/individual on others, however, it is unclear how this "influence" is measured. Data at hand does not appear to have inherent interactions among population members. Are authors using "interaction terms" (multiplicative terms) to measure influence? It is unclear. 

Response: Thank you for your comment. Yes, the authors use interaction terms (specifically, multiplicative terms) to measure the influence of one group or individual on others. These terms capture the interactions within the population, allowing for an analysis of how specific groups or individuals impact each other’s behaviours or outcomes.

2. Despite having many equations in the paper, the most important part is unexplained. How did the authors calculate coefficients? In Table 4, authors disaggregated the population according to their characteristics. In Table 5, authors appear to calculate coefficients for each of these groups. The problem is that, if you define the groups the way they did, there will be no variation in the independent variables. Eg: In case 1, everyone will have some education, poor and living in urban areas. This will cause perfect multicollinearity and we'll be unable to estimate coefficients. How did the authors come up with the estimates given in Table 5? 

Response: Thank you for your comment. The values assigned to the attributes in Table 4 for each of the eight cases describe the relevance of those attributes to the private incentive component of the Hamiltonian. For instance, in Case 1, having no formal education, being non-poor, and living in a rural region does not contribute to the private incentive of a group, whereas having some education, being poor, and living in an urban region does contribute to the private incentive of a group. In tables 5 we apply least squares method.

3. where are the estimates in Table 6 coming from? Do authors apply some sort of disaggregation in this estimation as well? it will be important to know how many observations are in each group.

Response: Thank you for your comment. In table 6, the interaction scenario, the independent variables are correlated. As a result, the least squares method is rendered ineffective. In that case, the partial least squares estimation method will be utilized. Kindly see lines 245 to 247.

4. In Table 2, No Education and Poor have negligible number of observations. Poor people are 5% of the rich ones and uneducated ones are 4% of the educated population. Can we get anything meaninful with such a huge difference? 

Response: Thank you for your comment. Despite the smaller sample sizes for "No Education" and "Poor" groups (4% and 5%, respectively, of their counterparts), the study's model can still provide meaningful insights. By applying statistical mechanics methods, specifically the Curie-Weiss model, the analysis accounts for population heterogeneity and uses both private and social incentives in decision-making. This approach allows for the evaluation of smaller group impacts within the broader socio-economic context, giving a reliable estimate of influence despite sample size disparities.

Reviewer 2 Comments 

1. This study explores the relationship between socio-economic factors and cooking fuel choices in Ghana, using data from the 2019 Ghana Malaria Indicator Survey (GMIS). The study highlights that this relationship becomes evident when social interactions are considered. However, several concerns arise. Although the topic is interesting and fills a gap in the literature on fuel use choices, the paper is written in overly technical language, making it difficult for readers to clearly understand the key findings, particularly regarding fuel choices among Ghanaian women of reproductive age. Simplifying the explanations would greatly improve accessibility and comprehension. Some sections, particularly in the discussions and conclusion, could benefit from further elaboration on policy implications or comparison with other studies in similar contexts outside Ghana. I have left my other comments in the attached pdf. The manuscript can be accepted for publications with some major changes. 

Response: Thank you for your thoughtful feedback. We acknowledge the need for a clearer explanation of our findings regarding cooking fuel choices among Ghanaian women. The revised manuscript simplifies technical language, particularly in the discussion and conclusion sections, and expands on policy implications by comparing similar studies from other regions. Kindly see lines 338-376 for the discussion section and lines 395-429 for the conclusion section.

2. Abstract: I suggest that these findings may be reported from LPG preference perspective first and then it's division of rural/urban and wealthier classification. To me it should first read 2.92% of educated women of reproductive age prefer LPG. Those who preferred non-LPM are 33.37 %. How about the remaining 59.93% of educated women of reproductive age? They didn't reveal their choice. 

Response: Thank you for your comment. I have updated the abstract in the revised manuscript. Kindly see lines 8-16 of the revised manuscript.

3. Methods: The methodology adopted does provide insights on the preferences of rural education women of reproductive age towards fuel choices, however the methods do not offer a causal interpretation of what drives them to these choices. I understand that the authors acknowledge this causality limitation in this study. With the current constraint, it thinks the authors can explore alternative models such as finding a good instrumental variable in explaining the education and fuel choice relationship using the same cross-sectional data. The paper lacks reference to seminal works in this area that follow the Curie-Weiss Model, and the methods adopted in this study. It is suggested that in the method section be enriched with recent works on similar area using this model.

Response: Thank you for your constructive feedback and insightful suggestions to improve our study. We appreciate your point on the limitations of our methodology in establishing causality. As you suggested, exploring the use of instrumental variables would be a valuable direction for future research, allowing us to strengthen causal inference in analyzing the relationship between education and fuel choice. Additionally, we recognize the importance of incorporating seminal works, particularly those that apply the Curie-Weiss Model, to situate our methodology within a broader theoretical framework. To enhance our study, we have added more applications of this model from key studies in related socio-economic contexts. These additions provide valuable perspectives on decision-making dynamics relevant to our research and enrich the methods section, aligning our approach more closely with established research paradigms. Kindly see lines 85-91 and lines 129-148.

4. Results:

The tables are not presented in the style of publication quality tables. A lot can be done with these tables to present in a more scientific way. Please follow empirical papers published in this journal and other journals.

Response: Thank you for your comment. I have revised the tables to meet publication standards by adopting a cleaner, more professional format and incorporating key elements from empirical journal tables. Each table now has a clear title and caption, making it easier to understand the data presented without additional context. Kindly see all tables of the revised manuscript.

5. Discussion: The idea that women of reproductive age in urban areas with some form of education show a significant preference for LPG does not bring the novelty in this area. It is almost a collective wisdom that education increase individuals’ awareness about environment and health. It would have been interesting to explore these women’s tendencies towards health (utilizing social interaction from health point of view). 

Response: Thank you for your comment. I have rewritten the discussion sections. Kindly see lines 338-375 of the revised manuscript.

6.This sentence seems a repetition of the previous sentence in this paragraph.

Response: Thank you for your comment. I have deleted the repeated sentence. 

7.These findings are not presented in reader-friendly way. I suggest that these findings may be reported from LPG preference perspective first and then it's division of rural/urban and wealthier classification. To me it should first read 2.92% of educated women of reproductive age prefer LPG. Those who preferred non-LPM are 33.37 %. How about the remaining 59.93% of educated women of reproductive age? They didn't reveal their choice?

Response: Thank you for your comment. I have updated the findings in the revised manuscript. Kindly see lines 8-16 and lines 258-266 of the revised manuscript.

---

## [Decision Letter · Decision Letter 1]

20 Dec 2024

Exploring Cooking Fuel Choices among Ghanaian Women of Reproductive Age: A Socio-Economic Analysis from a Statistical Mechanics Perspective

PONE-D-24-15863R1

Dear Dr. Ansah,

We’re pleased to inform you that your manuscript has been judged scientifically suitable for publication and will be formally accepted for publication once it meets all outstanding technical requirements.

Kind regards,

Muhammad Ali, PhD

Academic Editor

PLOS ONE

Additional Editor Comments (optional):

Reviewers' comments:

Reviewer's Responses to Questions

**Comments to the Author**

1. If the authors have adequately addressed your comments raised in a previous round of review and you feel that this manuscript is now acceptable for publication, you may indicate that here to bypass the “Comments to the Author” section, enter your conflict of interest statement in the “Confidential to Editor” section, and submit your "Accept" recommendation.

Reviewer #1: All comments have been addressed

2. Is the manuscript technically sound, and do the data support the conclusions?

Reviewer #1: Yes

3. Has the statistical analysis been performed appropriately and rigorously? 

Reviewer #1: Yes

4. Have the authors made all data underlying the findings in their manuscript fully available?

Reviewer #1: No

5. Is the manuscript presented in an intelligible fashion and written in standard English?

Reviewer #1: Yes

6. Review Comments to the Author

Reviewer #1: I have read the responses from the authors and I am satisfied with the incorporation of the comments. I suggest acceptance

7. PLOS authors have the option to publish the peer review history of their article (what does this mean?). If published, this will include your full peer review and any attached files.

Reviewer #1: No

---

## [Editor Report · Acceptance letter]

26 Dec 2024

PONE-D-24-15863R1 

PLOS ONE

Dear Dr. Ansah, 

I'm pleased to inform you that your manuscript has been deemed suitable for publication in PLOS ONE. Congratulations! Your manuscript is now being handed over to our production team.

Kind regards, 

on behalf of

Dr. Muhammad Ali 

Academic Editor

PLOS ONE